# Determinants of Influenza Vaccine Uptake Among Rural Populations in a Southeastern U.S. State

**DOI:** 10.3390/vaccines13121208

**Published:** 2025-11-29

**Authors:** Hanifat Hamzat, Oluchukwu M. Ezeala, Spencer H. Durham, Jingjing Qian, Salisa C. Westrick

**Affiliations:** 1Department of Health Outcomes Research and Policy, Auburn University Harrison College of Pharmacy, Auburn, AL 36849, USA; hzh0119@auburn.edu (H.H.); jzq0004@auburn.edu (J.Q.); 2Department of Pharmacy Practice, Auburn University Harrison College of Pharmacy, Auburn, AL 36849, USA; durhash@auburn.edu

**Keywords:** influenza, vaccinations, doctors trust, political affiliation, U.S.A., U.S., flu

## Abstract

Background and Objectives: Influenza is a significant global healthcare problem. Despite the availability of influenza vaccines, vaccination rates remain low, particularly among rural populations. This study aims to investigate the impact of trust and demographic factors on influenza vaccination status among rural populations. Methods: Cross-sectional data were collected as a subgroup analysis of rural populations within a parent study assessing Coronavirus Disease 2019 (COVID-19) vaccination uptake among residents in the state of Alabama, U.S.A. Participants were at least 18 years old and recruited from a Qualtrics panel. Rurality (non-metro) was determined using the United States Rural–urban Commuting Area (RUCA) Codes of 4-10. Data were analyzed using a weighted sample to adjust for differences in sex and race distributions. Results: A little over one-third (37.8%) of the participants reported having received influenza vaccine in the 2023–2024 flu season. Less than half (48.4%) reported they previously received COVID-19 vaccines, and a greater percentage (54.5%) of them had a high understanding of health information. The multivariable logistic regression analysis indicated that prior COVID-19 vaccination, political affiliation, household income and trust in doctors’ communication competency were significantly associated (*p* < 0.05) with greater influenza vaccine uptake. Conclusions: Improving influenza vaccine uptake in underserved rural communities requires strengthening trust in healthcare providers, addressing access barriers and enhancing communication strategies that reflect sociopolitical influences on vaccination behavior.

## 1. Introduction

Influenza remains a major global health issue, causing significant illness and death annually. During the 2023–2024 influenza season, the United States reported approximately 40 million influenza cases, 18 million healthcare visits, 470,000 hospitalizations, and 28,000 deaths [1]. Worldwide, seasonal influenza has led to an estimated 290,000–650,000 deaths annually [2].

Although vaccines have been available for over 60 years and have shown consistent safety and effectiveness [3] Influenza vaccine uptake in the United States remains suboptimal. During the 2023–2024 season, only 69.7% of older adults (aged 65 and above) and 55.4% of children (aged 6 months to 17 years) received the vaccine [4,5]. In Alabama, adult vaccination coverage was notably lower at 42.9%, placing the state among those with the lowest rates nationally [6]. Alabama’s persistently low vaccination rates, coupled with its large rural population and documented healthcare access challenges, make it a critical setting for investigating barriers to influenza vaccine uptake. The state’s unique combination of low coverage, racial disparities, and rural infrastructure gaps provides a compelling rationale for this study [7,8]. Additionally, there are ongoing racial and ethnic disparities; during the 2021–2022 season, vaccination rates among white adults were 53.9%, compared to only 42.0% of Black adults, 37.9% of Hispanic adults, and 40.9% of American Indian and Alaska Native adults [9].

Rural residents, including both adults and children, are 8 to 13 percent less likely to receive the influenza vaccine than their urban counterparts, reflecting consistent rural–urban disparities across age groups and sexes [10]. Contributing factors include disparities in trust toward the health system [11,12], time constraints, and limited access to vaccination services. However, trust in the health system among our study population in Alabama had not been previously assessed prior to this investigation. Rural communities face multiple barriers that complicate access to influenza vaccines. Geographic distance is a significant factor influencing vaccination rates; children whose parents live more than 10 miles from a healthcare provider are less likely to be vaccinated [10]. Rural health facilities frequently contend with staffing shortages, inadequate vaccine storage, and limited supply, which often force patients to seek vaccinations from government clinics or larger towns [12]. Additional challenges include lack of insurance or providers who do not accept it, transportation difficulties, work-related constraints, language barriers, and privacy concerns [13,14,15]. The shortage of healthcare workers, coupled with limited public transportation and internet access, exacerbates these issues and contributes to persistently lower vaccination rates in rural areas [12,16]. Further, the need for this research becomes evident with the decline in national influenza vaccination rates after the COVID-19 pandemic, which shows a troubling reversal of the progress made before the pandemic [17,18,19]. This decline in the vaccination rates requires prompt action and evidence-based strategies to prevent the loss of population immunity and avoid severe seasonal outbreaks that could overwhelm rural healthcare systems [20].

The trust placed in healthcare authorities and government health organizations directly impacts vaccine uptake [11,12]. As influenza vaccination rates continue to drop across all age groups, particularly in rural areas, there is an urgent need to increase vaccination coverage in these populations. Therefore, this research examines the influence of trust and demographic factors in the rural population regarding receipt of the influenza vaccine. This study aims to investigate the impact of trust and demographic factors on influenza vaccination status among rural populations in a southeastern U.S. state. By analyzing variables such as age, education, income, and perceived trust in healthcare providers, the study seeks to identify key determinants influencing vaccine uptake. These insights may inform targeted public health interventions to improve vaccination coverage in underserved rural communities.

## 2. Materials and Methods

### 2.1. Study Design, Sample and Setting

This cross-sectional study is an exploratory secondary analysis of data originally collected as part of the parent study assessing COVID-19 vaccination uptake among residents of the state of Alabama, U.S.A. [21]. The present analysis focused exclusively on participants residing in rural areas. Participants were at least 18 years old and recruited from a Qualtrics panel. Quota sampling was used to ensure diversity in ethnicity, race, COVID-19 vaccination status and residence in the original study. Rurality status (non-metro) was determined using the United States Rural–urban Commuting Area (RUCA) Codes of 4-10. This study was approved by the Institutional Review Board of the authors’ institution.

### 2.2. Data Collection and Measures

Data were collected from February to March 2024 using an online structured questionnaire. The survey sampling company, Qualtrics, was hired to handle the data collection process. A total of 424 participants out of the 1020 participants in the parent study who lived in rural areas were included. The flowchart for participant recruitment and inclusion is shown in Figure 1. The primary outcome was influenza vaccine uptake. Participants were asked if they received influenza shots during the 2023–2024 influenza season, with the response options being Yes, No, or Unsure. Only three participants selected Unsure and were excluded from the analysis.

The independent variables included participant characteristics and their trust in medical doctors, pharmacists and public health authorities. The characteristics assessed were sex, age, race, ethnicity, employment status, socio-economic status (household income and education), political affiliation, presence of co-morbidities or risk factors, confidence in understanding health information, and prior receipt of COVID-19 vaccination. Information on these variables was gathered through multiple-choice questions in the questionnaire (Appendix A).

Trust was measured using validated scales: Trust in Doctors in General (T-DiG), Trust in Community Pharmacists (TRUST-Ph), and Trust in Public Health Authorities (TiPHA). The T-DiG scale includes 29 items with seven factors: Communication Competency, Fidelity, Systems Trust, Confidentiality, Fairness, Stigma-Based Discrimination, and Global Trust. The TRUST-Ph scale consists of 30 items with three factors: Benevolence, Technical Competence, and Communication. TiPHA is a 14-item scale with two factors: Beneficence and Competence. All trust scales were evaluated using Likert scale ranging from ‘strongly disagree’ to ‘strongly agree.’ The TiPHA scale used a 4-point scale with no midpoint, while the T-DiG and TRUST-Ph scales used 5-point scales. The components within the three scales showed a Cronbach’s alpha greater than 0.70 using our dataset (Appendix B Table A1). All factors were retained and included in the final analysis.

### 2.3. Statistical Analysis

Primary data analysis was conducted using a weighted sample to adjust for differences between the sample and population sex and race distributions. Weighting was conducted by calculating the rural population proportions of four race–sex strata based on the 2023 Alabama population estimates provided by the U.S. Census Bureau (white/Male, Black and Other/Male, white/Female, Black and Other/Female) [22]. Each population proportion was then divided by the corresponding race–sex stratum sample proportion (Appendix B Table A2). The resulting weights were then assigned to participants based on their race–sex combinations. The weight distribution summary can be found in Appendix B Table A3.

Balance diagnostics were assessed by comparing the population versus unweighted sample, population versus weighted sample, and the weighted versus unweighted sample using absolute standardized mean differences (ASMDs) for sex and race. ASMDs were calculated using the following formula [23]:$$\mathrm{ASMD}=\frac{|p_{1}-p_{2}|}{\sqrt{\frac{p_{1}(1-p_{1})+p_{2}(1-p_{2})}{2}}}$$
where $p_{1}$ and $p_{2}$ represent the proportions in the reference and comparison groups, respectively. An ASMD of 0.1 was considered indicative of meaning imbalance [24]. The ASMDs (Appendix B Table A4) demonstrate that weighting improved balance between the sample and the population. Before weighting, sex distribution was strongly imbalanced (ASMD: population vs. unweighted = 0.48), with females overrepresented relative to the population (73.6% vs. 51.0%). However, after applying weights, the ASMD for population vs. weighted sample was 0.002, showing that the sample proportions closely match that of the population. Race distribution was relatively balanced in the unweighted sample (ASMD: population vs. unweighted = 0.031) and weighting further improved balance (ASMD: population vs. weighted sample = 0.017).

Frequencies and percentages were used to summarize participant characteristics and influenza vaccine uptake, while means and standard deviations were used to summarize the trust scales. Bivariate associations between the dependent and independent variables were assessed using Chi-square and binary logistic regression. Multicollinearity among the predictors that were significant in the bivariate analyses were assessed using linear regression. Appendix B Table A5 shows no significant multicollinearity was found (variance inflation factors > 5 were considered indicative of high multicollinearity) [25]. Multivariable logistic regression was then used to identify the most significant variables, out of those identified by the bivariate analysis, that were associated with influenza vaccine uptake. The model’s performance was assessed using several diagnostic checks. The area under the receiver operating characteristic curve (AUC) was calculated to assess how well the model discriminated between the binary outcome of influenza uptake (Yes vs. No). Internal validation was conducted using 1000 bootstrap resamples to provide a more robust estimate of performance. Additionally, the Box–Tidwell test was used to check the assumption of a linear relationship between the continuous predictors and the log odds of the outcome. All other data analyses conducted using the weighted sample were re-run with the unweighted data to assess the robustness of the study findings. The final model included all predictors in the multivariable analysis, regardless of statistical significance. No further modifications to model specifications, weighting procedures, or covariate selections were made after the analysis was finalized. Data analyses were performed using IBM SPSS Statistics, version 30.0.0.0, and R statistical software, version 4.2.1.

## 3. Results

### 3.1. Characteristics of the Study Participants

Weighted analyses were based on the total survey weights (*n* = 426). Table 1 shows that female and male participants were almost equally represented (50.9% vs. 49.1%). The majority were white (77.1%), and not Hispanic or Latino (94.6%). Most participants were aged between 35 and 64 years (55.3%), and 43.5% identified as Republicans. The most common educational attainment was a high school diploma or equivalent (54.2%). About 39% reported a household income of $30,000 or less, and 57.2% of the participants were retired, disabled or not employed. A large proportion (67.8%) reported having at least one chronic condition. Less than half (48.4%) of participants had previously received a COVID-19 vaccine. Lastly, over half (54.5%) reported high confidence in understanding health information.

### 3.2. Influenza Vaccine Uptake

When asked about the primary outcome of influenza vaccine uptake in 2023–2024, less than half (37.8%) reported receiving the influenza vaccine, as shown in Figure 2.

### 3.3. Associations Between Participant Characteristics and Influenza Vaccine Uptake in 2023–2024 Influenza Season

Table 2 presents the associations between participants’ sociodemographic factors and influenza vaccine uptake. Significant associations with influenza vaccine uptake (*p* < 0.05) were observed for sex, age, political affiliation, education, household income, employment status, presence of chronic conditions, and prior COVID-19 vaccination.

### 3.4. A Binary Logistic Regression Analysis of Participants’ Trust in Medical Doctors, Pharmacists, and Public Health Authorities on Influenza Vaccine Uptake

The binary logistic regression analyses (Table 3) revealed that all components of the trust in public health authority, medical doctor, and community pharmacist scales were significantly associated with increased odds of influenza vaccine uptake. Appendix A shows the results of the unweighted analysis. Bivariate results were largely consistent with the weighted analysis, with income not emerging as a significant factor (Appendix A). No evidence of multicollinearity was detected (Appendix A).

### 3.5. A Multivariable Logistic Regression of Factors Associated with Flu Vaccine Uptake

In the weighted multivariable logistic regression analysis (Table 4 and Figure 3), several factors remained significantly associated with the influenza vaccine uptake. First, previous COVID-19 vaccination status was the strongest predictor of influenza vaccine uptake. Participants who had previously received a COVID-19 vaccine were nearly 10 times more likely to receive the influenza vaccine than those who had not (adjusted OR = 9.79, 95% CI: 4.92–19.47, *p* < 0.001) Next, democrats had 2.5 times higher odds of vaccination than republicans (adjusted OR = 2.50, 95% CI: 1.02–6.16, *p* = 0.046). Participants with an income ≥ $120,000 had about 6.2 times higher odds of receiving the influenza vaccine compared to those with income ≤ $30,000 (adjusted OR = 6.17, 95% CI: 1.54–24.77, *p* = 0.010). Lastly, higher trust in doctors’ communication competency was significantly associated with greater influenza vaccine uptake (adjusted OR = 3.09, 95% CI: 1.56–6.12, *p* = 0.001). For each one-point increase in trust in doctors’ communication skills, participants’ odds of vaccination increased by 3.1 times.

For the unweighted analysis (Appendix A
Appendix A and Appendix A), only previous COVID-19 vaccination, political affiliation and trust in doctors’ communication competency remained significantly associated with influenza vaccine uptake. Participants who had previously received a COVID-19 vaccine were over seven times more likely to receive the influenza vaccine compared to those who had not (adjusted OR = 7.09, 95% CI: 3.88–12.95, *p* < 0.001). Next, compared to republicans, Independents had significantly higher odds of influenza vaccination (adjusted OR = 2.17, 95% CI: 1.02–4.61, *p* = 0.043). Lastly, higher trust in doctors’ communication competency was significantly associated with greater influenza vaccine uptake (adjusted OR = 2.65, 95% CI: 1.40–5.02, *p* = 0.003).

Marginal effects were estimated to examine the relationship between the Communication Competency component of the Trust in Doctors in General Scale and influenza vaccine uptake. Predicted probabilities were calculated across the full range of the communication competency scale (strongly disagree (1) to strongly agree (5)) and visualized using a marginal effects plot (Figure 4). The figure shows a positive association between Communication Competency and predicted probability of influenza vaccine uptake. As perceived communication competency score increased, predicted uptake also increased. This pattern was consistent in the unweighted analysis plot (Appendix A).

### 3.6. Model Diagnostics

The ROC curve (Appendix B Figure A1) with an area under the curve (AUC) of 0.8713 showed that the weighted model effectively distinguished between individuals who received the vaccine and those who did not. The ROC curve from the unweighted analysis also demonstrated good model discrimination, with an area under the curve (AUC) of 0.8710 (Appendix A). Internal validation of both models using nonparametric bootstrapping demonstrated strong stability. The weighted model correctly classified 83.4% of participants, with a minimal bias of 0.009 and a low standard error of 0.022. This finding is consistent with the unweighted analysis result in Appendix A. Linearity checks for the weighted model (Appendix B Table A6) and the unweighted model (Appendix A) showed that all continuous predictors had non-significant interactions with their log transformations (*p* > 0.05), satisfying the linearity assumption, except for Beneficence in the weighted model.

## 4. Discussion

This study focused on the factors revolving around the uptake of influenza vaccination among adults living in rural areas in Alabama during the 2023–2024 influenza season. Rural areas were studied because they have a comparatively lower vaccination uptake than urban centers [26] and have access to fewer healthcare facilities. Because of the low vaccination rates and limited availability of healthcare facilities, rural residents are vulnerable to influenza complications and fatalities if not treated promptly [27].

Trust in healthcare professionals emerged as one of the key factors in influenza vaccine uptake in our study, aligning with existing evidence that trust in providers and the healthcare system is a major determinant of vaccine acceptance and one of the strongest predictors of vaccination intent [11,28,29,30]. Similar patterns have been observed in other studies. For instance, Freimuth and colleagues [31] found that trust in the influenza vaccine significantly influenced uptake among African American and white adults, with lower trust contributing to disparities in immunization rates. Likewise, Viskupič and colleagues [32] demonstrated that trust in physicians was a statistically significant predictor of COVID-19 vaccine uptake among rural populations, reinforcing the role of provider trust across different vaccine contexts. These findings support our conclusion that trust in healthcare professionals, particularly in their communication competency, is a critical factor in vaccine acceptance, especially in underserved communities. Our study highlights a single component of trust in doctors in the communication competency domain. The importance of physicians’ communication skills extends beyond simply conveying information; it also encompasses delivering messages with empathy and cultural sensitivity. When providers communicate effectively, they are more likely to earn patients’ trust and address individual concerns. This trust, combined with strong and personalized recommendations [33,34], can significantly contribute to improving vaccination coverage. Provider-patient relationship is even more important in rural communities; evidence from rural populations shows that 86% of residents trust their personal healthcare providers for vaccine-related information, demonstrating the vital role of providers in these areas [35].

Prior COVID-19 vaccine uptake was the strongest independent predictor of influenza vaccine uptake in our study, which suggests that people who are open to one vaccine are more likely to accept others. This may be because of the general trust in vaccines or greater involvement in preventive health behaviors [36]. This trend aligns with recent studies indicating the COVID-19 pandemic has shaped attitudes toward other vaccines, both positively and negatively, based on individual experiences and trust in health authorities [12]. Therefore, we recommend that providers co-administer other needed vaccine(s) when patients present for a different vaccine [37]. It is recommended that healthcare providers should evaluate all vaccine needs during patient visits and offer appropriate vaccines at the same time when clinically appropriate. This method maximizes vaccination opportunities and helps ensure thorough immunization coverage. This is especially important among patients who may not be returning for follow-up visits [37].

Another finding from our study was the significant link between political affiliation and influenza vaccine uptake in the bivariate and multivariable logistic regression results. Compared with Republicans, Democrats (in the primary analysis) and Independents (in the unweighted analysis) had higher odds of being vaccinated against influenza. This finding aligns with recent studies showing that political partisanship is one of the strongest predictors of vaccine acceptance [38]. This divide appears to be extending beyond COVID-19 to include other vaccines, including the influenza vaccine [38,39]. The studies suggest that political identity shapes both vaccine attitudes and interpersonal influence, reinforcing the need for tailored messaging strategies. Pairwise contrasts between all political affiliation groups in both the primary (Table A7) and unweighted analyses (Appendix A) showed no statistically significant differences. This may be due to reduced power and adjustments for multiple comparisons. Given the inconsistencies in the results, the political affiliation findings should be interpreted with caution. Future research is recommended to further clarify the role of political affiliation in influenza vaccine uptake.

The primary analysis also identified income as a significant factor associated with influenza vaccine uptake. Specifically, individuals with high income (≥$120,000) had about 6.2 times higher odds of receiving the influenza vaccine compared to those with low income (≤$30,000). These results are consistent with prior research using National Health Interview Survey data from 2014–2018, which reported that adults with annual incomes of $100,000 or more were more likely to receive the influenza vaccine than those with family incomes below $35,000 [40]. Similarly, two additional studies examining income in relation to the federal poverty level found that individuals with higher income levels had increased odds of influenza vaccine uptake [41,42].

Although the influenza vaccine is widely available and safe, uptake in our study sample did not meet established public health goals, including the Healthy People 2030 target of 70% seasonal influenza vaccination coverage [43]. The findings highlight several important factors, suggesting that effective interventions will need a multi-pronged approach. This includes working with rural healthcare providers to improve communication between providers and patients. Strong vaccine recommendations should be delivered consistently, vaccines should be co-administered when clinically appropriate, and individual patient concerns should be addressed. Additionally, using trusted messengers within communities, such as local clinicians, religious leaders, and community influencers, may further improve outreach and engagement.

### Limitations

The cross-sectional design prevents causal inferences, and self-reported data may be affected by recall or social desirability bias. Although the sample is diverse, it may not fully represent all U.S. adults or those residing in urban areas. Also, the sociopolitical and demographic characteristics unique to Alabama, such as its partisan landscape and racial distribution, may not reflect those of other rural states. Additionally, because participants were recruited from a nonprobability online panel, external validity is limited even after weighting. Combining all non-white participants (including Black, Asian, American Indian or Alaskan Native, Multiracial and other) into a single category limited the ability to detect meaningful differences between specific racial groups. These subgroups were combined because the individual sample sizes were too small to allow reliable analysis. Next, our sample had a higher proportion of female participants compared to the target population, which further limits generalizability. However, the weighted analysis, which adjusted for both sex and race, showed that our findings were largely consistent with the unweighted and supports the robustness of the results. The measures of trust, while dependable, may not cover all aspects that are important to vaccine decision-making.

## 5. Conclusions

This study examined determinants of influenza vaccine uptake among rural populations in a southeastern U.S. state during the 2023–2024 influenza season. With only 37.8% of rural participants receiving the influenza vaccine, uptake remained well below public health targets. Four key factors emerged as significant predictors of vaccination: prior COVID-19 vaccination status, political affiliation, household income and trust in doctors’ communication competency.

These findings highlight the critical importance of trust-building between healthcare providers and rural patients, particularly through enhanced communication skills and routine vaccine recommendations. The strong association between COVID-19 and influenza vaccine uptake highlights opportunities for co-administration strategies to maximize vaccination coverage. Given the persistent rural–urban vaccination gap and declining post-pandemic influenza immunization rates, targeted interventions addressing trust, communication, and access barriers are urgently needed to improve influenza vaccine uptake in rural communities and protect population health.

## Figures and Tables

**Figure 1 vaccines-13-01208-f001:**
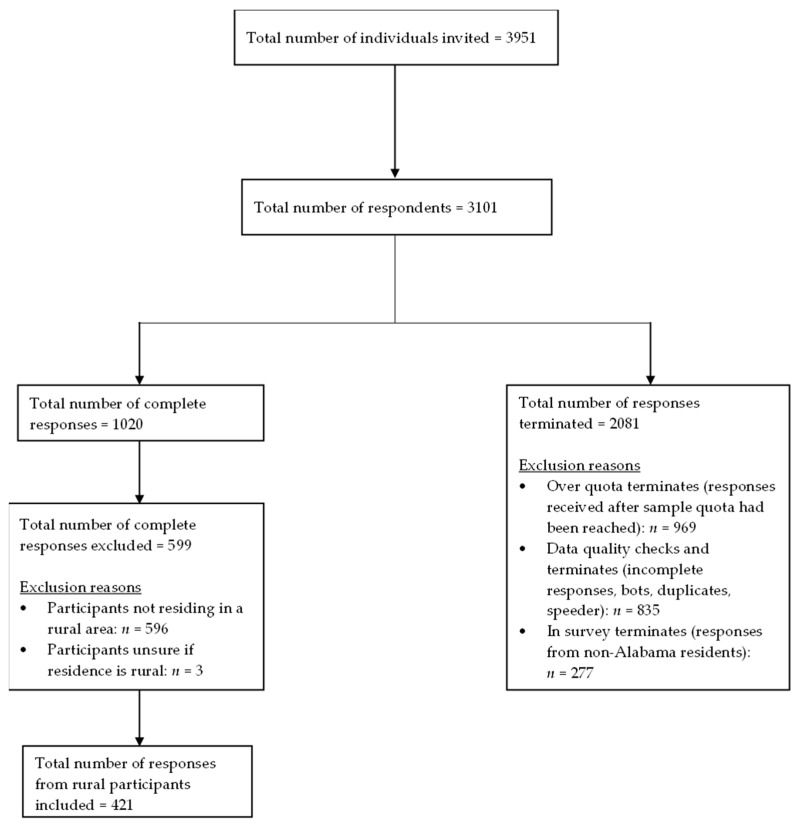
Flow diagram of participant recruitment and inclusion.

**Figure 2 vaccines-13-01208-f002:**
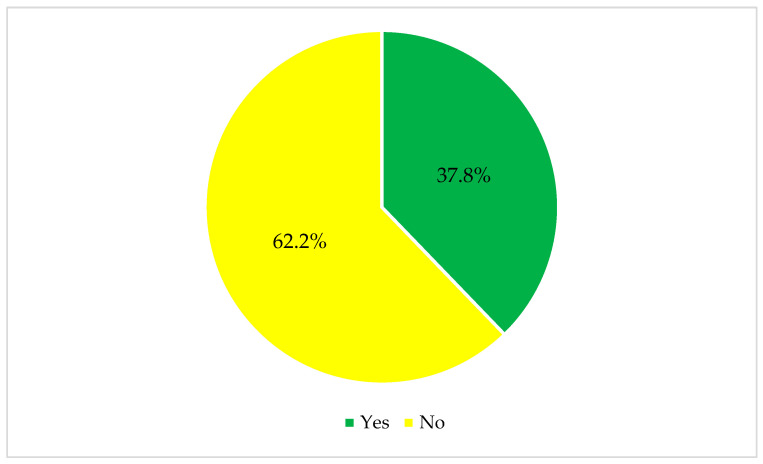
Influenza vaccine uptake (*n* = 426).

**Figure 3 vaccines-13-01208-f003:**
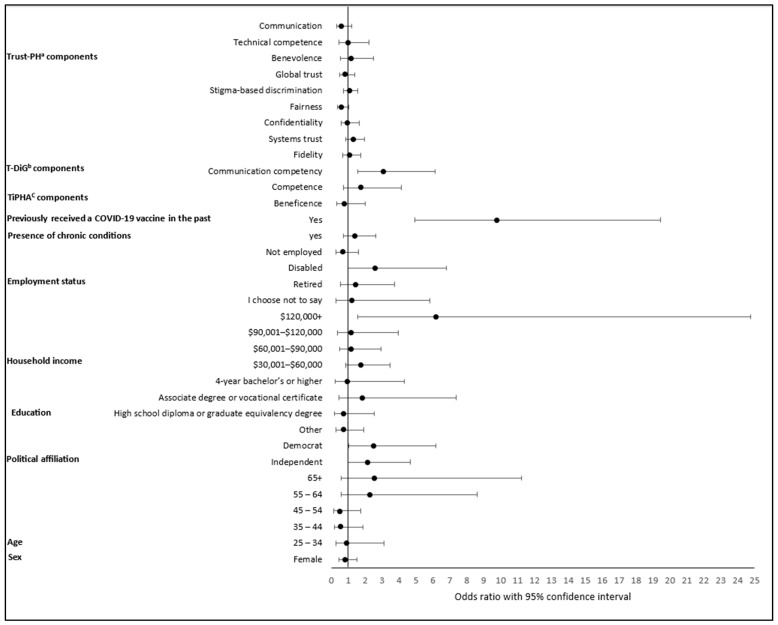
A multivariable logistic regression of factors associated with flu vaccine uptake (*n* = 426). ^a^ Trust in Community Pharmacists scale; ^b^ Trust in Doctors in General scale; ^c^ Trust in Public Health Authorities scale. Reference categories: Sex (male), age (18–24 years), political affiliation (republican), education (less than high school), income ($0–$30,000), employment status (employed), presence of chronic conditions (no), Previously received a COVID-19 vaccine in the past (no).

**Figure 4 vaccines-13-01208-f004:**
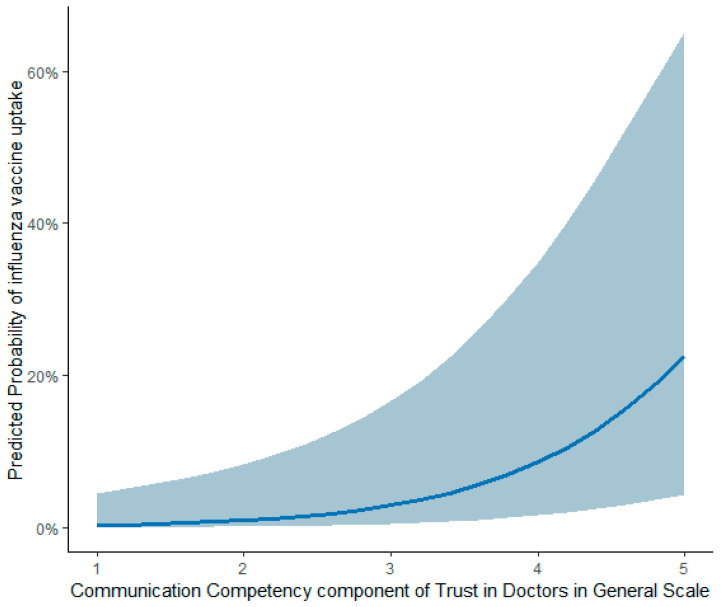
Marginal effects plot of communication competency on influenza vaccine uptake (*n* = 426).

**Table 1 vaccines-13-01208-t001:** Participant characteristics (*n* = 426).

Variable	*n* (%)
Sex	
Female	217 (50.9)
Male	209 (49.1)
Race	
White	328 (77.1)
Non-white *	98 (22.9)
Ethnicity	
Not Hispanic or Latino	403 (94.6)
Hispanic or Latino	23 (5.4)
Age	
18–24	38 (8.9)
25–34	68 (15.9)
35–44	87 (20.3)
45–54	84 (19.7)
55–64	65 (15.3)
65+	85 (19.9)
Political affiliation	
Republican	185 (43.5)
Independent	116 (27.3)
Democrat	58 (13.5)
Other	67 (15.7)
Education	
Less than high school	30 (7.1)
High school diploma or graduate equivalency degree	231 (54.2)
Associate degree or vocational certificate	107 (25.0)
4-year bachelor’s or higher	59 (13.7)
Household income	
$0–$30,000	165 (38.8)
$30,001–$60,000	129 (30.3)
$60,001–$90,000	71 (16.6)
$90,001–$120,000	25 (5.8)
$120,000+	19 (4.4)
I choose not to say	18 (4.1)
Employment status	
Employed	182 (42.8)
Retired	97 (22.7)
Disabled	59 (13.9)
Not employed	88 (20.6)
Presence of chronic conditions	
Yes	289 (67.8)
No	137 (32.2)
Previously received a COVID-19 vaccine in the past	
Yes	206 (48.4)
No	220 (51.6)
Confidence in understanding health information	
High	232 (54.5)
Moderate	140 (32.8)
Low	54 (12.8)

* Black: *n* = 69 (16.3%), American Indian/Alaska Native: *n* = 7 (1.6%), Asian: *n* = 1 (0.2%), Multiracial: *n* = 14 (3.2%) and Other (e.g., Spanish, Latina, Puerto Rico): *n* = 7 (1.7%).

**Table 2 vaccines-13-01208-t002:** Bivariate associations between participant characteristics and influenza vaccine uptake in 2023–2024 influenza season (*n* = 426).

Factor ^a^	Flu Vaccine Uptake ^a^		*p*-Value
	Yes *n* = 161 (37.8%) *n* (%)	No *n* = 265 (62.2%) *n* (%)	
Sex			0.020
Female	70 (43.5)	146 (55.1)	
Male	91 (56.5)	119 (44.9)	
Race			0.554
White	121 (75.6)	207 (78.1)	
Non-white ^b^	39 (24.4)	58 (21.9)	
Ethnicity			0.880
Not Hispanic or Latino	151 (94.4)	251 (94.7)	
Hispanic or Latino	9 (5.6)	14 (5.3)	
Age			<0.001
18–24	13 (8.1)	25 (9.4)	
25–34	19 (11.8)	49 (18.4)	
35–44	15 (9.3)	72 (27.1)	
45–54	23 (14.3)	61 (22.9)	
55–64	31 (19.3)	35 (13.2)	
65+	60 (37.3)	24 (9.0)	
Political affiliation			<0.001
Republican	67 (41.6)	118 (44.5)	
Independent	49 (30.4)	67 (25.3)	
Democrat	33(20.5)	25 (9.4)	
Other	12 (7.5)	55 (20.8)	
Education			<0.001
Less than high school	5 (3.1)	25 (9.4)	
High school diploma or graduate equivalency degree	64 (39.8)	167 (62.8)	
Associate degree or vocational certificate	60 (37.3)	47 (17.7)	
4-year bachelor’s or higher	32 (19.9)	27 (10.2)	
Household income			0.006
$0–$30,000	47 (29.2)	119 (44.7)	
$30,001–$60,000	53 (32.9)	76 (28.6)	
$60,001–$90,000	31 (19.3)	39 (14.7)	
$90,001–$120,000	13 (8.1)	12 (4.5)	
$120,000+	12 (7.5)	7 (2.6)	
I choose not to say	5 (3.1)	13 (4.9)	
Employment status ^a^			<0.001
Employed	60 (37.3)	122 (46.0)	
Retired	63 (39.1)	34 (12.8)	
Disabled	24 (14.9)	35 (13.2)	
Not employed	14 (8.7)	74 (27.9)	
Presence of chronic conditions			0.002
Yes	124 (77.0)	165 (62.3)	
No	37 (23.0)	100 (37.7)	
Previously received a COVID-19 vaccine in the past ^a^			<0.001
Yes	137 (85.1)	69 (26.0)	
No	24 (14.9)	196 (74.0)	
Confidence in understanding health information			0.973
High	88 (54.7)	144 (54.3)	
Moderate	52 (32.3)	88 (33.2)	
Low	21 (13.0)	33 (12.5)	

^a^ Chi square; ^b^ Influenza vaccine uptake for non-white participants: Black—Yes: *n* = 27 (16.9%), No: *n* = 42 (15.8%); American Indian/Alaska Native—Yes: *n* = 3 (1.9%), No: *n* = 4 (1.5%); Asian—Yes: *n* = 0 (0.0%), No: *n* = 1 (0.4%); Multiracial—Yes: *n* = 6 (3.8%), No: *n* = 7 (2.6%), Other (e.g Spanish, Latina, Puerto Rico)—Yes: *n* = 3 (1.9%), No: *n* = 4 (1.5%).

**Table 3 vaccines-13-01208-t003:** A binary logistic regression analysis of participants’ trust in medical doctors, pharmacists, and public health authorities on influenza vaccine uptake (*n* = 426).

Trust Scale	Odds Ratio ^a^	95% Confidence Interval		*p* Value
		Lower	Upper	
Trust in Public Health				
Beneficence	2.291	1.553	3.378	<0.001
Competence	2.200	1.544	3.133	<0.001
Trust in doctors in general				
Communication competency	2.444	1.781	3.352	<0.001
Fidelity	1.599	1.255	2.037	<0.001
Systems trust	1.414	1.140	1.753	0.002
Confidentiality	1.336	1.047	1.705	0.020
Fairness	1.381	1.083	1.761	0.009
Stigma-based discrimination	1.434	1.129	1.821	0.003
Global trust	1.841	1.442	2.351	<0.001
Trust in Community Pharmacists				
Benevolence	1.634	1.213	2.200	0.001
Technical competence	1.403	1.016	1.937	0.039
Communication	1.369	1.015	1.846	0.040

^a^ Reference category is no flu uptake.

**Table 4 vaccines-13-01208-t004:** A multivariable logistic regression of factors associated with flu vaccine uptake (*n* = 426).

Factor	Adjusted Odds Ratio ^a^	95% Confidence Interval		*p* Value
		Lower	Upper	
Sex (Ref = male)				
Female	0.809	0.438	1.493	0.498
Age (Ref = 18–24)				
25–34	0.901	0.260	3.124	0.869
35–44	0.551	0.161	1.883	0.342
45–54	0.525	0.158	1.751	0.295
55–64	2.258	0.593	8.596	0.233
65+	2.531	0.569	11.257	0.223
Political affiliation (Ref = republican)				
Independent	2.142	0.984	4.664	0.055
Democrat	2.504	1.017	6.164	0.046
Other	0.692	0.251	1.906	0.476
Education (Ref = less than high school)				
High school diploma or graduate equivalency degree	0.705	0.196	2.544	0.594
Associate degree or vocational certificate	1.837	0.458	7.370	0.391
4-year bachelor’s or higher	0.973	0.220	4.303	0.971
Income (Ref = $0–$30,000)				
$30,001–$60,000	1.723	0.854	3.474	0.128
$60,001–$90,000	1.194	0.489	2.916	0.697
$90,001–$120,000	1.164	0.342	3.964	0.809
$120,000+	6.172	1.538	24.769	0.010
I choose not to say	1.220	0.255	5.840	0.804
Employment (Ref = employed)				
Retired	1.429	0.548	3.727	0.466
Disabled	2.571	0.974	6.785	0.057
Not employed	0.672	0.285	1.582	0.362
Presence of chronic conditions (Ref = yes)				
no	1.370	0.711	2.643	0.347
Previously received a COVID-19 vaccine in the past (Ref = no)				
Yes	9.790	4.923	19.467	<0.001
Trust in Public Health				
Beneficence	0.782	0.303	2.019	0.611
Competence	1.722	0.714	4.154	0.226
Trust in doctors in general				
Communication competency	3.090	1.560	6.118	0.001
Fidelity	1.088	0.688	1.718	0.719
Systems trust	1.295	0.852	1.967	0.227
Confidentiality	0.973	0.578	1.638	0.919
Fairness	0.589	0.344	1.010	0.055
Stigma-based discrimination	1.046	0.701	1.561	0.824
Global trust	0.812	0.479	1.375	0.438
Trust in Community Pharmacists				
Benevolence	1.152	0.530	2.504	0.721
Technical competence	0.983	0.438	2.205	0.966
Communication	0.601	0.300	1.204	0.151

^a^ Reference category is no flu uptake.

## Data Availability

The study data is available upon reasonable request from the corresponding author.

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
