# Peer review of "Determinants of Influenza Vaccine Uptake Among Rural Populations in a Southeastern U.S. State"

_vaccines, 2025, doi:10.3390/vaccines13121208_

Round 1

Reviewer 1 Report

Comments and Suggestions for Authors

It was a pleasure to read the manuscript titled "Determinants of influenza vaccine uptake among rural populations in a southeastern U.S. state" submitted for publication in "Vaccines" journal.

Even if this paper follow a Cross-sectional approach focused in Alabama and specifically rural populations, I think that analyzing data regarding influenza vaccination is still very important given the fact that is a significant public health problem worldwide.

The paper was pleasant to read. Nonetheless, It is with some suggesstions that I recommend it for publication, here are my comments:

  1. In the abstract section, line 20, please add the full term with the acronym "Coronavirus Disease 2019 (COVID-19)" because it was the first time it appeared in the text.
  2. Please refer to the journal's instructions regarding the way that you must cite papers, and also names should be separated by commas, not semi-columns...etc.
  3. In the end of the introduction section, I think it would be better if you clearly emphasize the aim of the study in a specific paragraph, it will be clearer for the readers to directly identify why this study was conducted.
  4. Line 82, I don't understand, why did you put a citation here ?
  5. Line 86, please revise "authors’ institution’s Institutional Review Board" to avoid repetition.
  6. Line 112-119, please specify the software you used for statistical analyses
  7. The discussion section was very well written, I would just suggest to try to compare some findings with others from scientific litterature, you need also to cite more papers while doing that.
  8. Line 288, I think there are more abbreviations in the paper (eg. COVID-19, T-DiG, TRUST...etc.)

Apart from that, I have no more comments and I wish the authors good luck.

Reviewer 2 Report

Comments and Suggestions for Authors

Summary

This study investigates determinants of influenza vaccination among rural adults in Alabama, focusing on trust in health professionals, prior COVID-19 vaccination, and demographic factors.  The topic is timely and important.  However, several areas require substantial revision to strengthen methodological rigor and presentation.  I recommend major revisions and a subsequent round of review after the authors address the points below.

Comment:

  1. The manuscript relies heavily on tables. Several findings would be clearer as figures. For example, participant flow from parent study to rural analytic sample.

  1. I have some concerns related to the sample representativeness. The sample is imbalanced, such as 73.6% female, which limits generalizability.  I highly recommend conducting and reporting a post-stratification weighted sensitivity analysis: for age, sex, and race.  If weighting is not feasible, explicitly discuss the implications for external validity.

  1. Comments on the variable set.
    • In terms of political affiliation, the choice of reference category is not justified. Also, the group sizes are not reported.  Please provide cell counts, justify the reference category, and consider planned pairwise contrasts.
    • Collapsing to White vs non-White discards information and can mislead interpretation. I suggest using more granular categories when cell sizes allow. Otherwise, state clearly why this is not feasible and the limitations that follow.

  2. The exclusive focus on rural Alabama should be explicitly justified. Please explain why this setting was chosen and discuss how contextual features of Alabama may limit generalization to other rural regions.

  3. The authors need to provide a brief analytic plan summary and share data. The current version did not mention pre-registration or an analysis plan.  This can result as selection bias concerns. At least, I suggest authors clarify which analyses were pre-specified vs exploratory.

Round 2

Reviewer 2 Report

Comments and Suggestions for Authors

Overall, the authors have partially addressed the requested revisions. The flow diagram and the post-stratification sensitivity analysis are helpful. However, key items remain outstanding and should be addressed prior to acceptance:

  1. Resolve the Abstract inconsistency regarding the 35.2% figure.
  2. Add core figures (aOR forest plot; marginal-effects plot for the communication-trust scale).
  3. Report model diagnostics (AUC, calibration, internal validation) and linearity checks.
  4. Apply multiple-comparison control to political-affiliation contrasts and include results in the manuscript/supplement.
  5. Expand the weighting appendix (weight distribution, balance diagnostics, full weighted model tables).

Once these are addressed, the manuscript should be re-reviewed.

Author Response

Overall, the authors have partially addressed the requested revisions. The flow diagram and the post-stratification sensitivity analysis are helpful. However, key items remain outstanding and should be addressed prior to acceptance:

Authors: We apologize that the previous submission didn't fully address the reviewer's comments/suggestions. We hope that this version meets the expectations.

  1. Resolve the Abstract inconsistency regarding the 35.2% figure.

Response: We apologize for this oversight. The abstract has been updated to say “A little over one-third (35.2%) of the participants reported not having received influenza vaccine in the 2023-2024 flu season”.

  1. Add core figures (aOR forest plot; marginal-effects plot for the communication-trust scale).

Response: Thank you for your comment. Change has been as suggested. Please see Figures 3 and 4 in the manuscript text and Figures S2 and S3 in the supplementary file named S1 and S2.

  1. Report model diagnostics (AUC, calibration, internal validation) and linearity checks.

Response: Thank you. Changes have been made as suggested. Please see the Statistical Analysis section under Methods and Section 3.8 under Results.

  1. Apply multiple-comparison control to political-affiliation contrasts and include results in the manuscript/supplement.

Response: Thank you for your comment. Change has been as suggested. Please see Section 3.7 under Results. We also added a discussion point relevant to the results.

  1. Expand the weighting appendix (weight distribution, balance diagnostics, full weighted model tables).

Response: Thank you. We expanded the weighting appendix. See the Statistical Analysis section under Methods and Tables S1 to S3 in the supplementary file named S1 and S2.

Round 3

Reviewer 2 Report

Comments and Suggestions for Authors

<Summary>
I think the authors put some efforts. Presentation has improved, but key methodological issues remain that limit the strength of inference and generalizability.

<Major points>
Weighting and representativeness:
Please implement raking to age, sex, and race using appropriate state rural population margins. Report weight diagnostics: min, max, mean, SD, trimming rule, and effective sample size. Make the weighted model primary throughout and show unweighted only as sensitivity.

Model specification:
Replace bivariate screening with a theory-driven adjustment set or justify a penalized selection approach. If theory-driven, predefine age, sex, race, income, education, employment, chronic conditions, prior COVID-19 vaccination, political affiliation, and one trust construct.

Political-affiliation results:
Resolve the internal inconsistency. If pairwise comparisons do not survive multiplicity correction, revise the Abstract, Results, and Conclusions to report only the overall effect and move pairwise details to Supplement.

Trust constructs and collinearity:
Either prespecify one primary trust domain and treat others as secondary, or use dimension reduction or penalized regression. Show that results are stable to this choice.

Complete reporting:
Add a full adjusted odds-ratio table with CIs for all covariates in the primary model. Align figure captions, axis labels, and references. Fix remaining formatting artifacts.

Transparency:
Post a retrospective analysis note that freezes the final model, plus code and a de-identified dataset or synthetic data if needed. Update the Data Availability statement with links.

<Minor points>

If race must be collapsed for modeling, still provide descriptive counts and uptake for White, Black, and Other, and note this limitation explicitly.

Clarify that the sample is a nonprobability online panel and that external validity is limited even after weighting.

Author Response

Response to Reviewer’s Comments

I think the authors put some efforts. Presentation has improved, but key methodological issues remain that limit the strength of inference and generalizability.

  1. Weighting and representativeness:
    Please implement raking to age, sex, and race using appropriate state rural population margins. Report weight diagnostics: min, max, mean, SD, trimming rule, and effective sample size. Make the weighted model primary throughout and show unweighted only as sensitivity.

We appreciate your feedback.

As suggested, we have made the weighted model the primary analysis and included the unweighted model as a sensitivity analysis.

We implemented post-stratification weighting as suggested during Round 1 using rural Alabama population proportions for sex and race, as shown in Table A2 in the appendix. Age was not included in the weighting procedure because county-level rural population data were only available for sex and race.

The weight distribution and effective sample size are reported in Table A2 in the appendix. No trimming rule was applied because extreme values were not observed in the distribution, and the balance diagnostics (Table A3, Appendix section) showed that post-stratification weighting effectively adjusted the sample to align with rural Alabama population margins.

  1. Model specification:
    Replace bivariate screening with a theory-driven adjustment set or justify a penalized selection approach. If theory-driven, predefine age, sex, race, income, education, employment, chronic conditions, prior COVID-19 vaccination, political affiliation, and one trust construct.

We appreciate your feedback. This study was exploratory, and bivariate screening was used to identify which predictors were most likely to be relevant. All significant variables were subsequently assessed for multicollinearity using VIF, and retained predictors were entered into the final multivariable model. While not theory-driven, this approach is a commonly used approach and allowed us to identify meaningful associations and account for potential confounding while maintaining model stability.

  1. Political-affiliation results:
    Resolve the internal inconsistency. If pairwise comparisons do not survive multiplicity correction, revise the Abstract, Results, and Conclusions to report only the overall effect and move pairwise details to Supplement.

Thank you for your comment. The pairwise contrasts have been moved to Supplement. Discussion on the political affiliation finding has been toned down and recommendation for future studies suggested.

  1. Trust constructs and collinearity:
    Either prespecify one primary trust domain and treat others as secondary, or use dimension reduction or penalized regression. Show that results are stable to this choice.

Thank you for your comment. We conducted bivariate analyses between components of each of the three trust scales and the outcome to assess individual associations (Table 3). All significant components  were then evaluated for multicollinearity prior to inclusion in the multivariable model. No significant multicollinearity was identified (Table A5 in the Appendix), indicating that inclusion of all retained trust components did not compromise model stability.

  1. Complete reporting:
    Add a full adjusted odds-ratio table with CIs for all covariates in the primary model. Align figure captions, axis labels, and references. Fix remaining formatting artifacts.

Thank you for your comment. We originally had the table in the text, but we removed it given that the reviewer requested the figure in the previous round of review. In light of this round’s commend, as suggested, in addition to the figure, the full adjusted odds ratio table with confidence intervals for all covariates in the primary model has been readded (Table 4). Figure captions, labels, references and other formatting requirements have been aligned.

  1. Transparency:
    Post a retrospective analysis note that freezes the final model, plus code and a de-identified dataset or synthetic data if needed. Update the Data Availability statement with links.

Thank you for your comment. We have added the note on the final model in the last paragraph of the Statistical Analysis section. We have the Data Availability Statement to indicate that the data presented in this study are available upon request from the corresponding author.

  1. If race must be collapsed for modeling, still provide descriptive counts and uptake for White, Black, and Other, and note this limitation explicitly.

Thank you for your comment. We have added the descriptive counts and uptake as footnotes to Tables 1 and 2 in the manuscript and to Tables S2 and S3 in Supplementary File S2. The limitation of collapsing categories has been addressed in the Limitations section.

  1. Clarify that the sample is a nonprobability online panel and that external validity is limited even after weighting.

Thank you for your comment. Change has been made as suggested. Please see the Limitations section.